# Positive Patient Postoperative Outcomes with Pharmacotherapy: A Narrative Review including Perioperative-Specialty Pharmacist Interviews

**DOI:** 10.3390/jcm11195628

**Published:** 2022-09-24

**Authors:** Richard H. Parrish, Heather Monk Bodenstab, Dustin Carneal, Ryan M. Cassity, William E. Dager, Sara J. Hyland, Jenna K. Lovely, Alyssa Pollock, Tracy M. Sparkes, Siu-Fun Wong

**Affiliations:** 1Department of Biomedical Sciences, Mercer University School of Medicine, Columbus, GA 31902, USA; 2Department of Medical Science, Sobi Pharma, Waltham, MA 02451, USA; 3Department of Pharmacy Services, Crystal Clinic Orthopedic Center, Akron, OH 44312, USA; 4Department of Pharmacy Services, Mayo Clinic–Rochester, Rochester, MN 55905, USA; 5Department of Pharmacy Services, University of California Davis, Sacramento, CA 95817, USA; 6Department of Pharmacy Services, OhioHealth/Grant Medical Center, Columbus, OH 43215, USA; 7Department of Pharmacy Practice, Creighton University School of Pharmacy and Health Professions, Omaha, NE 68178, USA; 8Department of Pharmacy, University of Maryland Medical Center, Baltimore, MD 21201, USA; 9Division of Hematology/Oncology, Department of Medicine, UCI Health, Orange, CA 92868, USA

**Keywords:** clinical decision support systems, clinical pharmacology, collaborative practice, complications, Enhanced Recovery After Surgery, patient outcomes, perioperative care, pharmacotherapy, prophylaxis, risk assessment

## Abstract

The influence of pharmacotherapy regimens on surgical patient outcomes is increasingly appreciated in the era of enhanced recovery protocols and institutional focus on reducing postoperative complications. Specifics related to medication selection, dosing, frequency of administration, and duration of therapy are evolving to optimize pharmacotherapeutic regimens for many enhanced recovery protocolized elements. This review provides a summary of recent pharmacotherapeutic strategies, including those configured within electronic health record (EHR) applications and functionalities, that are associated with the minimization of the frequency and severity of postoperative complications (POCs), shortened hospital length of stay (LOS), reduced readmission rates, and cost or revenue impacts. Further, it will highlight preventive pharmacotherapy regimens that are correlated with improved patient preparation, especially those related to surgical site infection (SSI), venous thromboembolism (VTE), nausea and vomiting (PONV), postoperative ileus (POI), and emergence delirium (PoD) as well as less commonly encountered POCs such as acute kidney injury (AKI) and atrial fibrillation (AF). The importance of interprofessional collaboration in all periprocedural phases, focusing on medication management through shared responsibilities for drug therapy outcomes, will be emphasized. Finally, examples of collaborative care through shared mental models of drug stewardship and non-medical practice agreements to improve operative throughput, reduce operative stress, and increase patient satisfaction are illustrated.

## 1. Introduction

Enhanced postoperative recovery programs are evidence-based, multimodal, multidisciplinary approaches to the care of a surgical patient that involves a perioperative team aimed at reducing operative stress responses by increasing patient resilience, preventing and minimizing postoperative complications, and decreasing hospitalization [1,2]. The pharmacotherapy embedded into enhanced recovery programs and protocols (ERPs) has only recently received attention in the scientific and professional literature [3,4,5,6,7,8]. Often conducted outside of ERAS^®^ Society sites and protocols, many reports are retrospective and single-center in nature, and earlier reports evaluated programmatic compliance with consensus-generated guidelines and recommendations from various perioperative groups. Specifics related to medication selection, dosing, frequency of administration, and duration of therapy are evolving in a meaningful way to guide specific pharmacotherapeutic regimens towards agents of first choice for many enhanced recovery protocolized elements. This narrative review provides a summary of recent pharmacotherapeutic strategies, including those configured within electronic health record (EHR) applications and functionalities, that are associated with the minimization of the frequency and severity of postoperative complications (POCs), shortened hospital length of stay (LOS), reduced re-admission rates, and positive cost or revenue impacts. Further, it will highlight preventive pharmacotherapy regimens associated with improved patient preparation, especially those related to surgical site infection (SSI), venous thromboembolism (VTE), nausea and vomiting (PONV), postoperative ileus (POI), and delirium (PoD). The importance of interprofessional collaboration in all periprocedural phases, especially for medication management through shared responsibilities for drug therapy outcomes, will be emphasized. Finally, examples of collaborative care through shared mental models of drug stewardship programs and non-medical practice agreements to improve operative throughput, reduce operative stress, and increase patient satisfaction are illustrated in interviews with specialty perioperative clinical pharmacists.

## 2. Methods

A narrative review was undertaken following a PubMed and Google Scholar search for any English language report published in the last 10 years using the following primary search terms, separately and in combination: clinical decision support systems; clinical pharmacology; collaborative practice; complications; Enhanced Recovery After Surgery; patient outcomes; perioperative care; pharmacotherapy; pharmacist; prophylaxis; and risk assessment. Secondarily, individual medication names and ‘perioperative’ were also searched. Once identified, a snowball procedure was undertaken to identify and evaluate literature citations missed in PubMed and Google Scholar searches. The general research question was: what is the evidence for achievement of positive, postoperative patient outcomes with the use of pharmacotherapy and related functionalities?

Pharmacists interviewed were identified from the membership list serves of the American College of Clinical Pharmacy Perioperative Care and Critical Care Practice and Research Networks as well as those identified from a Google Scholar search for publication history and/or literature citations within perioperative care (see Appendix A. Each advanced practice or clinical pharmacist was interviewed through Zoom (San Jose, CA, USA, Version: 5.11.4 (7185)). The Zoom interviews were transcribed verbatim through its automated functionality, edited by RHP for sentence structure, timelines, and clarity, then returned to each interviewee for final editing. Interview questions are found in Table 1.

## 3. Results and Discussion

Three major themes were identified in the literature review and pharmacist interviews, including: (1) protocol development using computerized and manual care or order sets; (2) preventive measures associated with positive postoperative outcomes; and (3) interprofessional teamwork and collaborative practices among perioperative care members for achieving optimal pharmacotherapies.

### 3.1. Electronic Health Records (EHR) Systems and Perioperative Pharmacotherapy

One of the most significant challenges during ERP implementation is addressing the variability in drug regimens used within an organization throughout the perioperative period, that is, in pre-, intra-, and postoperative phases. An appreciation of the influence of pharmacotherapy on a patient’s recovery from surgery is beginning to emerge as measures to systematically follow care pathways via protocolized order sets and evaluate postoperative complication risks are more frequently applied in individual patient cases. Whether on paper or configured in EHR systems, these protocols often are operationalized as preprinted or preformatted care or order sets that attempt to guide pharmacotherapy decision-making toward evidence-based regimens and selections. While many inpatient institutions worldwide continue to use paper order sheets for the transmission of medical orders to various departments, preprinted order sets compared to handwritten orders have been shown to reduce errors and patient morbidity. In a study of hand-written compared to preprinted standard care sets, over 80% of cases utilizing handwritten orders had at least one omission error compared to 38% in the group that used preprinted sets. Further subgroup analyses demonstrated that errors in mechanical VTE prophylaxis and SSI prophylaxis orders were significantly reduced in the preprinted group compared to the handwritten group [9]. Some reports have described the establishment of a specialty-specific, mandatory, computerized clinical decision support (CDS) module for VTE risk stratification and prevention. Application of so-called “hard-stop” or “forced-function” rules has been shown to improve best practice VTE prophylaxis from 51.1% to 97.8%, a 2-fold improvement attained through multidisciplinary teamwork [10]. After implementation of an evidence-based, specialty-specific “smart order set”, risk-appropriate VTE prophylaxis prescriptions increased significantly from 65.6% to 90.1%, and orders for any form of VTE prophylaxis increased from 76.4% to 95.6%. The rate of radiographically documented symptomatic VTE within 90 days of hospital discharge declined from 2.5% to 0.7%. Preventable harm from VTE was completely eliminated with no difference in major bleeding or all-cause mortality [11]. Computer-based order intervention significantly improved the proportion of surgeries with timely discontinuation of antibiotics from 38.8% to 55.7% [12].

The development of order sets that serve to operationalize and standardize enhanced recovery elements at pre-, intra- and postoperative phases has been described in a recent report [13]. In addition to the use of CDS risk assessments, among their recommendations for safer surgery include: (1) clear identification of the patient’s ERAS status in the EHR, particularly on the patient’s electronic chart header and the computer screen banner on summary/overview page; (2) creation and use of an ERAS column on perioperative status board; (3) adoption of a daily order set progression with separate pre-op, intra-op, postop phase sets; and (4) incorporation of a safe surgery checklist. Table 2 shows an example of a preoperative antibiotic prophylaxis order set for colorectal procedures. Many preoperative risk assessments have been integrated into EHR computerized provider order entry (CPOE) systems as a part of an initial work-up. Table 3 illustrates the scope and variety of CDS systems for predicting and preventing POCs that can be integrated into CPOE systems or used as stand-alone tools.

### 3.2. Preventive Measures and Methods Associated with Positive Postoperative Outcomes

One of the most effective ways to minimize surgical risk from medications is to identify, resolve, and prevent drug therapy problems (DTPs) [15,16,17]. The purpose of identifying, preventing, and/or resolving DTPs is to understand a patient’s drug-related needs and to help them achieve their treatment goals and realize the best possible outcomes from drug therapy [18]. Risk factors associated with the incidence of DTPs have included polypharmacy (≥5 scheduled medications daily), drug allergies, BMI > 25 kg/m^2^, and creatinine clearance < 30 mL/min [19]. In addition, recent, current, or chronic use of any of the following medications and substances increases the likelihood of an adverse outcome in any patient during the perioperative period, including corticosteroids, opioids, insulin, anticoagulants, proton pump inhibitors, cancer chemotherapy, immunomodulators, ethanol, tobacco, and illicit substances [20]. Institutions have implemented DTP-related process and practice changes at key times in the admission process, that is, six to eight weeks preoperatively (prehospital) to detect and manage anemia, hyperglycemia, and smoking [21], in the preadmission clinic [22,23,24], on admission during medication reconciliation [25,26,27,28], throughout hospital stay [29,30], and at discharge [31,32]. In addition, theater and ward-based activities to rationalize drug therapy [33,34] include identifying and avoiding or minimizing the use of potentially inappropriate medications for at-risk patient populations: (1) the American Geriatric Society’s (AGS) Beers list for older adults [35]; and (2) the KIDs list for children created by the Pediatric Pharmacy Association (PPA) [36]. Table 4 outlines the medications and classes on the AGS Beers and PPA KIDs lists that should be avoided in perioperative care patients due to delirium-producing, falls-risk, reduced renal elimination, or coagulation adverse effects.

Within ERPs, prevention of POCs is paramount to patient outcomes [37,38,39,40]. Commonly targeted POCs have included SSI, VTE, PONV, POI, and PoD. For example, prevention of SSI with standardized intravenous and oral antibiotics; VTE through combinations of pharmacologic agents and non-pharmacologic devices; PONV and POI through goal-directed fluid therapy (GDFT), early feeding, opioid reduction, and peripherally acting µ receptor antagonist (PAMORA) use; and acute kidney injury (AKI) with GDFT and by judicious use and dosing of NSAIDs have become recognized best practices to minimize these complications, improve throughput, and facilitate earlier hospital discharge [41].

Several medications have been developed to address common scenarios in perioperative care that often delay hospital discharge. However, many enhanced recovery reports describing the benefits of high cost medications, such as parenteral acetaminophen and NSAIDs like parenteral meloxicam, and alvimopan, topical liposomal bupivacaine (alone and combined with meloxicam), sugammadex, and dexmedetomidine are limited because the overall costs of the intervention are not included, and cost savings associated with avoidance of negative outcomes are not monetized in the analyses. Moreover, not all of these agents are available in every country nor on the drug formulary of every hospital. Furthermore, none of the above agents are listed in the 2021 World Health Organization model list of essential drugs, and access is very likely to be non-existent in low- to middle-income countries (LMICs) [42,43]. ERP concentration centers on maintenance of or return to normal gastrointestinal functioning through early oral intake and route switching, including nutrients and medications. A summary of those medications mentioned in the above paragraph follows.

On balance, acetaminophen is best used orally on a scheduled basis and within a multi-modal pain management regimen that includes intrathecal hydromorphone, a scheduled oral NSAID, and adjuvant preoperative gabapentinoid with intraoperative IV magnesium, dexmedetomidine, and/or lidocaine infusions [44]. However, large randomized trials are needed before IV lidocaine becomes standard of practice [45]. The increased cost of IV acetaminophen is not offset by any additional benefit in pain management, LOS, or readmission [46,47,48,49,50,51,52,53,54,55,56,57]. With concomitant opioid treatment, alvimopan use may be associated with shorter times to tolerance of a soft diet, return of gastrointestinal function, and decreased LOS. However, no randomized clinical or cost-effectiveness trials have validated these findings in ERAS^®^ patients [58,59,60,61], and other safer, less costly agents exist, such as naloxegol and naldemedine [62,63]. Nevertheless, a 5-month retrospective pilot study to reduce alvimopan use after GI function recovery successfully eliminated over two-thirds of postoperative doses through pharmacist recommendation to discontinue [64]. Dexmedetomidine (now available as a multi-source generic) has been shown to facilitate extubation for earlier PACU discharge [65,66] and mitigate postpartum depression [67], but, when combined with ketamine as a continuous infusion, may not yield lower opioid consumption or pain scores [68]. Sugammadex may shorten time to extubation and may accelerate bowel function recovery and shorter postoperative LOS as compared to reversal of neuromuscular blockade with neostigmine and glycopyrrolate [69,70]. For use as a local anesthetic by wound infiltration, liposomal bupivacaine (LB) may reduce opioid use and hospital LOS as well as promote earlier oral diet [71,72,73], but this effect may be procedure-specific [74]. Currently, while evidence exists for the use of IV ketorolac, a non-selective cyclooxygenase (COX) enzyme inhibitor, as a safe and effective component of multimodal analgesia [75,76], data are emerging to support the use of IV and topically applied meloxicam in postoperative medication management based on its longer half-life and cyclooxygenase-2 only inhibition [77,78]. Additionally, use of metamizole (dipyrone) as a unique analgesic with a mechanism mediated through non-COX enzymatic inhibition of prostaglandin E2 formation, is widely employed in Europe for postoperative pain [79,80,81]. Limited evidence from a recent Cochrane review of its oral use as a single dose in acute postoperative pain demonstrated efficacy but without any comparators [82]. While unavailable in Canada, United States, Australia, Japan and the Middle East and not included on the WHO List of Essential Drugs, metamizole can be considered as an evolving short-term nonopioid analgesic in other parts of the world. While intrathecal opioid administration may be gaining in popularity over epidural and IV administration (TIVA) in visceral procedures, it remains controversial and may be associated with increased hospital costs due to prolonged stay in intensive care [83,84,85,86,87,88].

Mechanical bowel preparation (MBP) with laxatives for bowel cleaning and antibiotics for gut sterilization as part of an SSI regimen may not improve overall patient outcomes in colorectal surgery (i.e., lower SSI rate and LOS), and may precipitate postoperative fluid (up to 2 L preoperative loss) and electrolyte imbalances [89,90]. Its value in non-colorectal procedures is limited. However, laparoscopic procedures and those requiring intraoperative colonoscopy to localize a lesion that cannot otherwise be identified may be facilitated using MBP [91].

It has been estimated that between 15 and 20% of surgical patients have undiagnosed diabetes or impaired fasting glucose which does not necessarily affect postoperative outcomes [92,93] unless the plasma glucose level was ≥250 mg/dL, which was associated with higher 30-day morbidity/mortality [94]. Glucose management in patients with and without diabetes could be impacted by oral carbohydrate and dexamethasone administration. In patients without diabetes, positive patient effects for both mothers (lower insulin level and HOMA-IR) and neonates (higher blood glucose) have been observed. Preoperative carbohydrate loading may improve insulin resistance and reduce PONV and hospital LOS [95,96]. In general, single-dose dexamethasone up to 10 mg raises blood glucose with similar preoperative to maximal intraoperative glucose concentration of 63 ± 69 mg/dL in diabetics and to 72 ± 45 mg/dL in nondiabetics [97,98,99,100,101]. Preoperative hyperglycemia is associated with increased SSI risk, among others, especially in the elderly [102,103].

Another programmatic systems approach to medication management endeavors to create a shared mental model using drug stewardship as a platform of organizational change [104,105,106,107]. While most stewardship programs have concentrated on rational antimicrobial use [108], recent developments have focused attention on medication classes that are frequently used in perioperative care and are high-risk for the patient, including opioids [109,110,111,112,113], anticoagulants [114,115,116,117,118,119,120], anemia management [121,122,123,124,125], and glycemic control with insulin [126,127], among others [4]. Table 5 summarizes recommendations for preventing VTE, SSI, PONV, POI, and PoD [128]. Cefazolin is the only injectable first-generation cephalosporin available in most countries and should be specified by name in ERPs; other injectable first generation cephalosporins, cephalothin and cephapirin, are no longer available for use. Available injectable second-generation cephalosporins include cefuroxime, cefoxitin, and cefotetan; cefamandole has been discontinued.

In the near future, pharmacogenomic testing results (from either blood or saliva) can provide guidance on how to optimize pharmacotherapy for each patient based on an individual’s unique genetic profile. Genetics-guided pharmacotherapy and its impact on clinical outcomes needs to be thoroughly studied for better understanding and managing drug administration in ERAS settings [129]. For example, pharmacogenomic testing to detect CYP3A5 single nucleotide polymorphism (SNP) is used to guide tacrolimus dosing in solid organ transplantation. Evidence exists for the CYP2D6 SNP related to codeine and tramadol ultrarapid metabolism and life-threatening cardiovascular and respiratory depression as well as CYP2C9 testing for celecoxib and warfarin to prevent bleeding [130]. Carriers of the OPRM A118G allele may have higher postoperative opioid requirements [131,132].

In general, the use of pharmacotherapy throughout the perioperative process is highly variable in regimen selection, dosing, frequency, and duration. As illustrated in Appendix A, variation in one relatively simple regimen, VTE prophylaxis specificity, from one excellent meta-analysis of hepatic resections leads to differences in VTE occurrences [140]. Efforts to reduce this process variation, both at the same institution and among institutions, are underway at many facilities and organizations to optimize the contribution of pharmacotherapy to patient outcomes.

### 3.3. Interprofessional Collaboration and Teamwork among Perioperative Disciplines

For other common surgical complications, protocols for mitigating each complication have been developed and implemented through interdisciplinary collaboration often through creation of care bundles [1,2,3,4,141,142]. One recent review noted that multidisciplinary surgical care often leads to better patient outcomes and improved provider knowledge as well as directly leading to cost savings, irrespective of surgical specialty, modality, or intervention [143]. As noted earlier, VTE prophylaxis standards have been embedded into order sets and monitoring plans [144], and surgery discharge plans have been designed and adopted [145]. A postoperative atrial fibrillation pathway has been developed, implemented, and successfully sustained that identifies and correct the underlying arrhythmic cause, as well as choosing a strategy for rate or rhythm control and determining thrombotic risk [146]. Prevention of PONV has been addressed through multidisciplinary patient-centered initiatives by implementing multi-modal strategies that capitalize on timely administration of agents with different mechanisms of action [147,148]. Three excellent examples of interprofessional collaboration and shared mental modeling for perioperative medication management applicable in ERPs include the online UK Handbook of Perioperative Medicines [149], a manual on decision making in perioperative care [150], and a handbook covering enhanced recovery optimization [151]. Standards of practice for pharmacy services in perioperative medicine as well as inpatient quality measures for clinical pharmacist practice have been derived recently [4,152,153,154].

Navigating toward successful collaborative care practices is often dependent on the hospital or health system’s jurisdiction, policies and procedures regarding the level of collaboration, such as institution-wide, service-specific, procedural, and individual practitioner scopes of practice, and local credentialing and privileging history and mechanisms. Thus, the division of labor for perioperative drug therapy management in the United States is often dispersed among licensed independent providers (surgeons and anesthesiologists), mid-level providers (nurse practitioners and physician assistants), and advanced practice or clinical pharmacists. In some state jurisdictions, clinical pharmacists are considered healthcare providers. The Centers for Disease Control and Prevention has published an evidentiary document for establishing, implementing, and sustaining collaborative care best practices and comprehensive drug therapy management through formal CPAs [155]. The American Society of Health-system Pharmacists (ASHP) has provided a link to state-specific CPAs [156]. Table 6 summarizes the “practice pearls” unique to the pharmacotherapy of each surgical specialty area.

Pharmacists attached to various surgical services have provided consultation to anesthesiologists at the service and individual patient levels regarding local anesthetic systemic toxicity (LAST) and enhanced recovery analgesic/anesthetic pathways (in light of the recent move toward increased local anesthetic use to reduce opioid exposure) as well as evolving surgical techniques. Collaboration with surgeons has often focused on risk assessments and treatment recommendations for preoperative and pharmacokinetic-based antimicrobial optimization [157], VTE and PONV prophylaxis [144,147], postoperative pain management, POI mitigation, and electrolyte replacement and nutrition support, including glycemic control [158]. In one 20-month pre-/post-study in which clinical pharmacists were directly involved, pre-implementation colonic SSIs were reduced from 10% to 2%, VTE rate decreased from 0.6% to 0%, and postoperative readmission rate decreased from 4.8% to 1.3% [111].

In conclusion, best practices in perioperative-related pharmacotherapies are emerging as complementary interventions via use of protocolized order sets to improve patient resilience to common POCs. Further, employment of risk assessments in the preoperative phase assists in identifying potentially problem-prone scenarios. Identification, prevention, and resolution of drug therapy problems, including medications to avoid in certain patient subgroups, combined with organized drug stewardship programs for opioids, antibiotics, and anticoagulants, among others, forms the basis of pharmaceutical care in the perioperative setting. Numerous examples exist to illustrate how interprofessional teamwork and collaborative practice structures and standards facilitate interventions improve patient outcomes [159,160,161,162,163,164]. In Appendix A, perioperative-specialty pharmacist practice examples with literature citations that improve patient outcomes through comprehensive medication management and identify both facilitators and barriers through drug stewardship programs are described [165,166,167,168,169,170,171,172,173,174,175,176,177,178,179,180,181,182,183,184,185,186,187,188,189,190,191,192,193,194,195,196,197,198,199,200,201,202,203,204,205,206,207,208,209,210,211,212,213,214,215,216,217,218,219,220,221,222].

## Figures and Tables

**Table 1 jcm-11-05628-t001:** Questions about Collaborative Care Practices Used for Perioperative Advanced Practice or Clinical Pharmacist Interviews (see Appendix A).

How did perioperative collaborative care practices begin at your institution? Where did it start? How did it diffuse to other areas or service lines? At what point was clinical pharmacy incorporated into the collaborative care model? What’s your collaborative care practice story? How have drug stewardship programs evolved at your institution? What medication classes are included? In what physical areas, surgical service lines, or clinical functions does clinical pharmacy have responsibility and accountability for medication management for perioperative patients? ◯ PAC, OR, PACU, ward, etc.; ortho, general, thoracic, bariatric, etc.; type of non-medical prescribing—institutional protocol-based, individual CPAs, independent, supplemental, etc. What methods were effective in sustaining collaborative care? Pertaining to enhanced recovery, what metrics or measurements are used to assess practice effectiveness? Are there any specific metrics related to pharmacotherapy or medication management?

**Table 2 jcm-11-05628-t002:** Preoperative Antibiotic SSI Prophylaxis Order Set for ERAS^®^ Elective Small Bowel and Colorectal Procedures. (Note: “look-alike, sound-alike” medication names with TALLMAN letters to reduce medication error [14]).

**ERAS Colorectal Surgery, Adult—Inpatient Pre-Op Order Set** Antibiotic Prophylaxis Antibiotics should be given within 60 min prior to incision. **For Elective Small Intestine, Non-obstructed procedures:** Choose ONE option: Option 1 □ ceFAZolin 2 g IV once pre-operatively If patient has ceFAZolin allergy or severe non-IgE mediated reaction to any β-lactam: Option 2 □ gentamicin (1.5 mg/kg) _________ mg IV once pre-operatively AND □ clindamycin 600 mg IV once pre-operatively **For Elective Colorectal and Anal procedures:** Choose ONE option: Option 1 □ ceFAZolin 2 g IV once pre-operatively AND □ metroNIDAZOLE 500 mg IV once pre-operatively If patient has ceFAZolin allergy or severe non-IgE mediated reaction to any β-lactam: Option 2 □ gentamicin (1.5 mg/kg) _________ mg IV once pre-operatively AND □ clindamycin 600 mg IV once pre-operatively If patient has ceFAZolin allergy or severe non-IgE mediated reaction to any β-lactam: Option 3 □ gentamicin (1.5 mg/kg) _________ mg IV once pre-operatively AND □ metroNIDAZOLE 500 mg IV once pre-operatively

**Table 3 jcm-11-05628-t003:** Examples of Risk Assessment Tools Available for CPOE Systems as Clinical Decision Support (CDS).

Operative Complication	Risk Assessment Tool	Website
General surgical risk of complication	American College of Surgeons NSQIP Surgical Risk Calculator	https://riskcalculator.facs.org/RiskCalculator/ (accessed on 30 May 2022)
General preoperative patient health	American Society of Anesthesiologists Physical Status Classification System	https://www.mdcalc.com/asa-physical-status-asa-classification#evidence (accessed on 30 May 2022)
Ethanol withdrawal	Clinical Institute Withdrawal Assessment for Alcohol scale—revised (CIWA-Ar)	https://www.merckmanuals.com/professional/multimedia/clinical-calculator/ciwa-ar-clinical-institute-withdrawal-assessment-for-alcohol-scale (accessed on 30 May 2022)
Venous thromboembolism (VTE)	Caprini Score for Venous Thromboembolism (2005)	https://www.mdcalc.com/caprini-score-venous-thromboembolism-2005 (accessed on 30 May 2022)
Surgical site infection (SSI)	SSI Risk Index	http://www.ohri.ca/SSI_risk_index/Default.aspx (accessed on 30 May 2022)
Nausea and vomiting (PONV)	Apfel Score for Postoperative Nausea and Vomiting	https://www.mdcalc.com/apfel-score-postoperative-nausea-vomiting (accessed on 30 May 2022)
Acute kidney injury (AKI)	RIFLE Criteria for Acute Kidney Injury	https://www.mdcalc.com/rifle-criteria-acute-kidney-injury-aki (accessed on 30 May 2022)
Postoperative ileus (POI)	Charlson Comorbidity Index	https://www.mdcalc.com/charlson-comorbidity-index-cci (accessed on 30 May 2022)
Postoperative hyperglycemia	American Diabetes Association risk calculator	https://www.mdcalc.com/american-diabetes-association-ada-risk-calculator (accessed on 30 May 2022)
Postoperative delirium (PoD)	4 A’s Test for Delirium Screening	https://www.mdcalc.com/4-test-4at-delirium-assessment (accessed on 30 May 2022)

**Table 4 jcm-11-05628-t004:** AGS Beers and KIDs Listed Medications and Classes with Greater Risk to Benefit.

AGS Beers List	PPA KIDs List
Medications with high anticholinergic burden: first generation histamine-1 antagonists, antispasmodics, tricyclic antidepressants, and most antipsychotics in patients with Parkinson disease complicated by psychosis, although quetiapine, clozapine, and pimavanserin may be used with caution	Codeine and tramadol in children unless pharmacogenetic testing is used
Rivaroxaban and dabigatran in older adults because of a higher bleeding risk than warfarin and other direct oral anticoagulants	Meperidine in neonates and caution in children due to risk for respiratory depression due to active metabolite
Tramadol due to risk of hyponatremia from syndrome of inappropriate antidiuretic hormone secretion	Midazolam in very low birth weight neonates
Opioids with benzodiazepines or gabapentinoids (gabapentin, pregabalin) because the combinations increase the risk of severe respiratory depression	Ceftriaxone with caution in neonates due to formation of kernicterus
Nonsteroidal anti-inflammatories (indomethacin, celecoxib, ketorolac, naproxen, etc.)	Mineral oil in neonates and infants due to lipid pneumonia
Non-benzodiazepine hypnotics (zolpidem, zopiclone, eszopiclone) due to hangover effects and falls risk	Opium tincture and paregoric in neonates and children due to respiratory depression, gasping syndrome, seizures, CNS depression, and hypoglycemia
Certain cardiovascular medications: amiodarone, spironolactone, calcium channel blockers	Sodium phosphate solution, rectal (enema) in infants due to electrolyte abnormalities, acute kidney injury, arrhythmia, and death
Meperidine due to risk for delirium and neurotoxicity	Propofol in doses greater than 4 mg/kg/h for more than 48 h due to propofol-related infusion syndrome; higher rate in children than adults because higher relative doses of propofol are needed, especially in status epilepticus
Estrogens and testosterone due to cardiovascular or carcinogenic issues	Dopamine antagonists used as anti-emetics due to acute dystonia (dyskinesia); increased risk of respiratory depression, extravasation, and death with INTRAVENOUS use: prochlorperazine, haloperidol, metoclopramide, promethazine, and trimethobenzamide

**Table 5 jcm-11-05628-t005:** Summary of Pharmacotherapy Recommendations for Preventing Common POCs.

Postoperative Complication	Recommendations (Note: Alternatives Are Needed in an Era of Drug Shortages)
Venous thromboembolism (VTE) [133,134]	Any appropriately dosed and timed LMWH (>12 h after neuraxial anesthesia) ◯ LMWHs—better outcomes than w/unfractionated heparin ◯ BID LMWH associated w/incr. risk of spinal hematoma Avoid rivaroxaban and dabigatran in the elderly due to increased bleed risk compared to warfarin Continue for 28 d in cancer patients/risk stratify TJA patients for appropriate agent selection (LMWH, DOAC, ASA) and duration
Surgical site infection (SSI) (one pre-op dose; discontinue within 24 h) [135,136]	Intravenous antibiotics are part of a bundled approach to SSI prevention that includes normothermia maintenance, perioperative glucose control, appropriate hair removal, oral antibiotic bowel preparation (laparoscopic procedures), preoperative bathing w/ chlorhexidine, standardized postoperative dressing removal/wound care, and wound closure protocol including glove w/or w/o gown change/separate instrument tray. Oral antibiotic gut sterilization has little value outside of laparoscopic bowel and rectal procedures. Clean (high-risk)/clean-contaminated procedure (SSI risk—1–8%) ◯ **Adults**—cefazolin 2 g IV to start ≤120 min prior to incision (alternative—cefuroxime 1.5 g IV)—broadened time window especially helpful when using vancomycin (15 mg/kg) if MRSA colonized ◯ **Children**—cefazolin (or cefuroxime) 50 mg/kg IV Contaminated procedure (SSI risk—20–25%) ◯ **Adults**—cefazolin w/metronidazole 500 mg IVPB (except w/recent EtOH consumption due to “disulfiram” reaction—increased PONV) ◯ **Children**—cefazolin w/metronidazole 7.5 mg/kg IV ◯ Alternatives—cefoxitin OR cefotetan 2 g IV OR ertapenem 1 g IV (all cover anaerobes) All cephalosporins and ertapenem can be given IV push diluted with 20 mL over 3–5 min; metronidazole 5 mg/mL IVPB over 20–60 min Allergy to cephalosporins ◯ **Adults**—gentamicin 1.5 mg/kg IVPB w/metronidazole IVPB 500 mg OR clindamycin 600–900 mg IVPB—infusion needs to begin at least 60 min pre-operatively ◯ **Children**—gentamicin w/metronidazole as above OR clindamycin 10 mg/kg IVPB
Nausea and vomiting (PONV) [137]	Preoperative complex carbohydrate loading PONV prophylaxis—multimodal approach ▪ Preoperative ▪ aprepitant 1–3 h prior for ≥2 risk factors ▪ Muscarinic antagonists (scopolamine patch) ▪ Order to leave on skin behind ear for 72 h ▪ Intraoperative ▪ dexamethasone 8–10 mg IV (half-life 36–54 h) ▪ Postoperative (around the clock for 48 h postop) ▪ 5HT_3_I (ondansetron, granisetron, polonosetron) ▪ Dopamine (D_2_) antagonists (metoclopramide, droperidol, prochlorperazine); metoclopramide (prokinetic) may aid gut peristalsis ▪ Histamine-1 antagonists (diphenhydramine, dimenhydrinate, trimethobenzamide)
Ileus (POI) [138]	bisacodyl 5–10 mg orally twice daily beginning POD-1 magnesium hydroxide (MOM) 30 mL—separate administration time from bisacodyl due to bisacodyl’s enteric coating PAMORAs for opioid-containing regimens—no CNS penetration ▪ May improve postop GI recovery/reduce LOS ▪ alvimopan 12 mg oral if taking opioids ▪ ≤15 doses only due to increased risk of MI; stop upon passing flatus ▪ Safer PAMORAs w/evidence of effectiveness in preventing POI for opioid-induced constipation in chronic pain ▪ naloxegol 12.5–25 mg oral ▪ naldemedine 0.2 mg oral
Delirium (PoD) [139]	Avoid prolonged (>6 h) fluid fasting w/goal-directed fluid therapy Offer water and/or clear liquids until 1–2 h preoperatively Comprehensive geriatric assessment, including pharmacotherapy Use of multimodal opioid-sparing analgesia—acetaminophen and COX-2 NSAIDs (i.e., celecoxib, meloxicam); account for current opioid exposure, i.e., chronic pain Consider use of intraoperative IV dexmedetomidine and ketamine Avoid intraoperative benzodiazepines and gabapentinoids

**Table 6 jcm-11-05628-t006:** Pharmacotherapy “practice pearls” for each specialty area.

Surgical Specialty Area	Practice Pearls
General perioperative	Begin collaborative care for medication management with one surgical team or service ◯ Branch out to pre-admission, PACU, ward, and discharge phases Focus on programmatic antimicrobial, anticoagulant, and opioid stewardship Participate in order/care set/protocol development Develop multidisciplinary pathways to mitigate AKI, delirium, and atrial fibrillation Measure outcomes in terms of POC, LOS, and readmission reductions Estimate cost reduction impacts and/or revenue optimization Develop drug shortage mitigation plans for perioperative pharmacotherapy
Bariatrics	Concentrate on postoperative dose formulation management Use COX-2 inhibitors (celecoxib, meloxicam) for pain to avoid anastomotic leaks
Cardiothoracic	Be mindful of any hardware insertion that makes the patient prone to bleeding Modify medication doses based on constant assessment of renal function
Colorectal	Develop multidisciplinary NSAID use criteria to minimize AKI Be vigilant to change medication routes to oral to promote gut function return
Gynecological oncology	Provide supportive care for PONV and pain management Develop collaborative care for initiation and monitoring of oral chemotherapy
Orthopedics	Lead rehabilitation medication education Develop both institution-based and individual CPAs
Pediatrics	Actively question the need for venous access to prevent CLABSIs in neonates Develop interdisciplinary medication weaning procedures for opioids/benzodiazepines
Solid organ transplant	Focus on medication management/optimization for immunosuppressive regimens Prepare living donor medication management plans ahead of the transplant
Vascular	Develop interdisciplinary blood pressure augmentation protocols and MAP goals Be vigilant for and manage AKI due to lower limb ischemia during aneurism repairs

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
