# Peer review of "Positive Patient Postoperative Outcomes with Pharmacotherapy: A Narrative Review including Perioperative-Specialty Pharmacist Interviews"

_jcm, 2022, doi:10.3390/jcm11195628_

Round 1
Reviewer 1 Report
Please consider including a methods section that highlights approach to manuscript identification and selection and rationale for the selection of examples provided (section 5). In addition, consider shortening the interviews and/or reducing the number of interviews, as there is some redundancy within and between interviews.
Extensive narrative review of enhanced recovery programs and protocols. in addition, a large section of the manuscript is devoted to interviews with pharmacists who have experience designing, implementing, and utilizing such programs/protocols.
The literature review is thorough, and the topic is of interest. The interviews (section 5) are a bit lengthy and somewhat redundant.
My main concern however is the lack of any methods section that explains the choice of studies quoted in the narrative review, and the approach used to identify interviewees and conduct interviews.
Author Response
jcm-1907200 – author responses in italics
Reviewer 1:
Please consider including a methods section that highlights approach to manuscript identification and selection and rationale for the selection of examples provided (section 5). In addition, consider shortening the interviews and/or reducing the number of interviews, as there is some redundancy within and between interviews.
We appreciate the reviewer’s comment about the lack of structure, and have restructured the manuscript using IMRD.
Extensive narrative review of enhanced recovery programs and protocols. in addition, a large section of the manuscript is devoted to interviews with pharmacists who have experience designing, implementing, and utilizing such programs/protocols.
We agree with the reviewer’s insightful comment about the interview section and have made section 5 into a supplement. The discussion section has summarized the similarities and differences between the different perioperative specialties.
The literature review is thorough, and the topic is of interest. The interviews (section 5) are a bit lengthy and somewhat redundant.
We thank the reviewer for this observation and have restructured the interviews as a supplement.
My main concern however is the lack of any methods section that explains the choice of studies quoted in the narrative review, and the approach used to identify interviewees and conduct interviews.
We are grateful for this comment and have included a methods section that (1) details the search strategy and (2) outlines the approach for identifying and interviewing clinical pharmacists.
Reviewer 2 Report
The authors have written an extensive and detailed narrative review about the influence of pharmacotherapy regimens on surgical patient outcomes I think the topic is relevant for JCM.
Major comments
- The article is too extensive and therefore not easy to read. The reason for this is mainly because of the detailled interviews in the second part of the manuscript. Although they are interesting to read, they could be provided as a a supplement so that the main text is more concise and to the point. The authors could provide some practical recommendations per area instead of the whole interview. -
- I miss the discussion and conclusion..
Minor comments
1. Introduction
Line 1. The authors should provide a definition of enhanced recovery programs and protocol (ERPs).
3. Preventive measures and methods associated with positive postoperative outcomes
Line 161. Intrathecal hydromorphone is quite something in my opinion and not a drug/route of administration you will use regularly in a multimodal pain management regimen. For which specific cases do the authors recommend this?
Metamizol is, at least in Europe, becoming more and more popular during the perioperative period since it has less contra-indications/side-effects compared to the classical NSAIDs. Maybe the authors can elaborate in this.
https://databankws.lareb.nl/Downloads/2017_2_metamizole_analgesia_nsaids_Pain_Practice.pdf
Table 3. Delirium/PONV: important factors to avoid these is drinking water/clear liquids until 1-2 hours before surgery.
Table 4. In order to keep the article concise and to the point, I don't think that this table really adds much. It is sufficient to mention the text.
5.
Line 323. What was the effect of your intervention on the SSI rate?
Although I think that the role of the pharmacist is important the optimize pharmacotherapy during the peri-operative case, it is the main expertise of anesthesiologists. How do you authors see this collaboration? Where begins the responsibility of the anesthesiologist/pharmacist and where does it end?
I miss the discussion and conclusion..
Author Response
jcm-1903200 – author responses in italics
Reviewer 2:
The authors have written an extensive and detailed narrative review about the influence of pharmacotherapy regimens on surgical patient outcomes I think the topic is relevant for JCM.
We thank the reviewer for this comment.
Major comments
- The article is too extensive and therefore not easy to read. The reason for this is mainly because of the detailled interviews in the second part of the manuscript. Although they are interesting to read, they could be provided as a a supplement so that the main text is more concise and to the point. The authors could provide some practical recommendations per area instead of the whole interview. –
We appreciate the reviewer’s comment and have made the interview section as a supplement.
- I miss the discussion and conclusion.
We thank the reviewer for this comment and have restructured the manuscript using IMRD format.
Minor comments
- Introduction
Line 1. The authors should provide a definition of enhanced recovery programs and protocol (ERPs).
We are grateful for the reviewer’s observation and have added a definition for ERPs with two recent and salient citations from Kehlet [1] and Ljungqvist [2] in the introduction.
Enhanced postoperative recovery programs are evidence-based, multimodal, multidisciplinary approaches to the care of a surgical patient that involves a perioperative team aimed at reducing operative stress responses by increasing patient resilience, preventing and minimizing postoperative complications, and decreasing hospitalization.
Kehlet, H. Enhanced postoperative recovery: good from afar, but far from good? Anaesthesia. 2020, 75, e54-e61. doi: 10.1111/anae.14860.
Ljungqvist, O.; Scott, M., Fearon, K.C. Enhanced Recovery After Surgery: A Review. JAMA Surg. 2017, 152, 292–298. doi:10.1001/jamasurg.2016.4952
- Preventive measures and methods associated with positive postoperative outcomes
Line 161. Intrathecal hydromorphone is quite something in my opinion and not a drug/route of administration you will use regularly in a multimodal pain management regimen. For which specific cases do the authors recommend this?
We thank the reviewer for this comment. Intrathecal narcotic administration is not “state of the art” for most procedures and is controversial. Therefore, other options for pain management, such as epidural and blocks will be added.
Metamizol is, at least in Europe, becoming more and more popular during the perioperative period since it has less contra-indications/side-effects compared to the classical NSAIDs. Maybe the authors can elaborate in this.
https://databankws.lareb.nl/Downloads/2017_2_metamizole_analgesia_nsaids_Pain_Practice.pdf
We appreciate the reviewer’s comment on metamizole as a unique pain reliever. We have added the following paragraph:
"Additionally, use of metamizole (dipyrone) as a unique analgesic with a mechanism mediated through non-COX enzymatic inhibition of prostaglandin E2 formation, is widely employed in Europe for postoperative pain [77,78,79]. Limited evidence from a recent Cochrane review of its oral use in single-dose acute postoperative pain demonstrated efficacy but without any comparators [80]. While unavailable in Canada, United States, Australia, Japan and the Middle East and not included on the WHO List of Essential Drugs, metamizole can be considered as an evolving short-term nonopioid analgesic in other parts of the world."
We added the following references:
Cascorbi, I. The Uncertainties of Metamizole Use. Clin. Pharmacol. Ther. 2021, 109, 1373-1375. doi: 10.1002/cpt.2258.
Lutz, M. Metamizole (Dipyrone) and the Liver: A Review of the Literature. J. Clin. Pharmacol. 2019, 59, 1433-1442. doi: 10.1002/jcph.1512.
Konijnenbelt-Peters, J.; van der Heijden, C.; Ekhart, C.; Bos, J.; Bruhn, J.; Kramers, C. Metamizole (Dipyrone) as an Alternative Agent in Postoperative Analgesia in Patients with Contraindications for Nonsteroidal Anti-Inflammatory Drugs. Pain Pract. 2017, 17, 402-408. doi: 10.1111/papr.12467.
Hearn, L.; Derry, S.; Moore, R.A. Single dose dipyrone (metamizole) for acute postoperative pain in adults. Cochrane Database Syst. Rev. 2016, 4, CD011421. doi: 10.1002/14651858.CD011421.pub2.
Table 3. Delirium/PONV: important factors to avoid these is drinking water/clear liquids until 1-2 hours before surgery.
We appreciate the reviewer’s comment and have added this clinical pearl to the table.
Table 4. In order to keep the article concise and to the point, I don't think that this table really adds much. It is sufficient to mention the text.
We agree with the reviewer and will mention it in the text.
Line 323. What was the effect of your intervention on the SSI rate?
We agree with the reviewer and have added the change in SSI rate at line 330.
Although I think that the role of the pharmacist is important the optimize pharmacotherapy during the peri-operative case, it is the main expertise of anesthesiologists. How do you authors see this collaboration? Where begins the responsibility of the anesthesiologist/pharmacist and where does it end?
We fully agree with the reviewer about the collaboration between anesthesiologists and clinical pharmacists. We have added two paragraphs in the discussion regarding areas of primary and secondary focus for pharmacotherapy management between surgeons, anesthesiologists, and clinical pharmacists in pre-, intra-, and postoperative phases of care. At line 309,
“Navigating toward successful collaborative care practices is often dependent on the hospital or health system’s jurisdiction, policies and procedures regarding the level of collaboration, such as institution-wide, service-specific, procedural, and individual practitioner scopes of practice, and local credentialing and privileging history and mechanisms. Thus, the division of labor for perioperative drug therapy management in the United States is often dispersed among licensed independent providers (surgeons and anesthesiologists), mid-level providers (nurse practitioners and physician assistants), and advanced practice or clinical pharmacists. In some state jurisdictions, clinical pharmacists are considered healthcare providers. The Centers for Disease Control and Prevention has published an evidentiary document for establishing, implementing, and sustaining collaborative care best practices and comprehensive drug therapy management through formal CPAs [155]. The American Society of Health-system Pharmacists (ASHP) has provided a link to state-specific CPAs [156].
Pharmacists attached to various surgical services have provided consultation to anesthesiologists at the service and individual patient levels regarding local anesthetic systemic toxicity (LAST) and enhanced recovery analgesic/anesthetic pathways in light of the recent move toward increased local anesthetic use to reduce opioid exposure as well as evolving surgical techniques. Collaboration with surgeons has often focused on risk assessments and treatment recommendations for preoperative and pharmacokinetic-based antimicrobial optimization [157], VTE and PONV prophylaxis [144,147], postoperative pain management, POI mitigation, and electrolyte replacement and nutrition support, including glycemic control [158]. In one 20-month pre-/post-study in which clinical pharmacists were directly involved, pre-implementation colonic SSIs were reduced from 10% to 2%, VTE rate decreased from 0.6% to 0%, and postoperative readmission rate decreased from 4.8% to 1.3% [111].”
I miss the discussion and conclusion.
We appreciate this comment and have restructured the manuscript to include a discussion and conclusion.
Round 2
Reviewer 2 Report
I am satisfied with the answer. Some last comments:
- 'Pharmacists interviewed were identified from the membership listserv of '. Typo.
- What kind of method was used to analyse the results of the interviews?
- 'Avoid drinking water and/or clear liquids until 1-2 hours preoperatively.' Driking should NOT be avoided.
- As a reader, I would find it interesting to get some practical tips from the interviews without having to read everything. Maybe the authors could give a short overview of practical tips per specialy area.
Author Response
jcm-1907200 - author responses in italics
Reviewer 2 - second review
I am satisfied with the answer. Some last comments:
We thank the reviewer for the insightful comments.
- 'Pharmacists interviewed were identified from the membership listserv of '. Typo.
We will correct the typo. Thank you.
- What kind of method was used to analyse the results of the interviews?
We appreciate this comment. There was no analysis of the transcripts per se. "We added the following statement in the Methods section: "The Zoom interviews were transcribed verbatim through its automated functionality, edited by RHP for sentence structure, timelines, and clarity, then returned to each interviewee for final editing."
- 'Avoid drinking water and/or clear liquids until 1-2 hours preoperatively.' Driking should NOT be avoided.
We understand what the reviewer was addressing and will change the statement to read: "Offer water or clear liquids until 1-2 hours preoperatively."
- As a reader, I would find it interesting to get some practical tips from the interviews without having to read everything. Maybe the authors could give a short overview of practical tips per specialy area.
We appreciate this comment and have added a "practice pearls" table in the main text to outline tips for the area.
|
Surgical specialty area |
Practice pearls |
|
General perioperative |
· Begin collaborative care for medication management with one surgical team or service o Branch out to pre-admission, PACU, ward, and discharge phases · Focus on programmatic antimicrobial, anticoagulant, and opioid stewardship · Participate in order / care set / protocol development · Develop multidisciplinary pathways to mitigate AKI, delirium, and atrial fibrillation · Measure outcomes in terms of POC, LOS, and readmission reductions · Estimate cost reduction impacts and/or revenue optimization · Develop drug shortage mitigation plans for perioperative pharmacotherapy |
|
Bariatrics |
· Concentrate on postoperative dose formulation management · Use COX-2 inhibitors (celecoxib, meloxicam) for pain to avoid anastomotic leaks |
|
Cardiothoracic |
· Be mindful of any hardware insertion that makes the patient prone to bleeding · Modify medication doses based on constant assessment of renal function |
|
Colorectal |
· Develop multidisciplinary NSAID use criteria to minimize AKI · Be vigilant to change medication routes to oral to promote gut function return |
|
Gynecological oncology |
· Provide supportive care for PONV and pain management · Develop collaborative care for initiation and monitoring of oral chemotherapy |
|
Orthopedics |
· Lead rehabilitation medication education · Develop both institution-based and individual CPAs |
|
Pediatrics |
· Actively question the need for venous access to prevent CLABSIs in neonates · Develop interdisciplinary medication weaning procedures for opioids/benzodiazepines |
|
Solid organ transplant |
· Focus on medication management/optimization for immunosuppressive regimens · Prepare living donor medication management plans ahead of the transplant |
|
Vascular |
· Develop interdisciplinary blood pressure augmentation protocols and MAP goals · Be vigilant for and manage AKI due to lower limb ischemia during aneurism repairs |
|
Table 5: Pharmacotherapy “practice pearls” for each specialty area |
|
